# Partial Disturbance of Microprocessor Function in Human Stem Cells Carrying a Heterozygous Mutation in the DGCR8 Gene

**DOI:** 10.3390/genes13111925

**Published:** 2022-10-23

**Authors:** Dóra Reé, Ábel Fóthi, Nóra Varga, Orsolya Kolacsek, Tamás I. Orbán, Ágota Apáti

**Affiliations:** 1Institute of Enzymology, Research Center for Natural Sciences, 1117 Budapest, Hungary; 2Doctoral School of Molecular Medicine, Semmelweis University, 1085 Budapest, Hungary

**Keywords:** miRNA processing, C19MC, miRNA cluster, human pluripotent stem cells, differentiation

## Abstract

Maturation of microRNAs (miRNAs) begins by the “Microprocessor” complex, containing the Drosha endonuclease and its partner protein, "DiGeorge Syndrome Critical Region 8" (DGCR8). Although the main function of the two proteins is to coordinate the first step of precursor miRNAs formation, several studies revealed their miRNA-independent functions in other RNA-related pathways (e.g., in snoRNA decay) or, for the DGCR8, the role in tissue development. To investigate the specific roles of DGCR8 in various cellular pathways, we previously established a human embryonic stem-cell (hESC) line carrying a monoallelic DGCR8 mutation by using the CRISPR-Cas9 system. In this study, we genetically characterized single-cell originated progenies of the cell line and showed that DGCR8 heterozygous mutation results in only a modest effect on the mRNA level but a significant decrease at the protein level. Self-renewal and trilineage differentiation capacity of these hESCs were not affected by the mutation. However, partial disturbance of the Microprocessor function could be revealed in pri-miRNA processing along the human chromosome 19 miRNA cluster in several clones. With all these studies, we can demonstrate that the mutant hESC line is a good model to study not only miRNA-related but also other “noncanonical” functions of the DGCR8 protein.

## 1. Introduction

MicroRNAs (miRNAs) are small noncoding RNAs, representing an abundant class of key regulatory molecules that act on the posttranscriptional level of gene expression. They function in cytoplasmic ribonucleoprotein complexes that always contain a member of the Argonaute (Ago) protein family. These RNA-induced silencing complexes (RISCs) constantly scan the RNA population of the cell to find their targets by base pair complementarity between the contained miRNA and other, mainly protein coding mRNAs (in animals, the target sites are mostly located at the 3′ untranslated regions). The effector functions of miRNAs are manifested in either translation inhibition or the destabilization and degradation of the target mRNAs. As miRNAs generally have several target molecules, and mRNAs usually contain numerous miRNA binding sites, this posttranscriptional fine-tuning represents an elaborate regulatory network with a complexity comparable with that of transcription factors [1,2,3]. This explains why miRNAs play important regulatory roles in most cellular processes, having essential functions especially during embryonic development. Imbalance of the miRNA proportions is widely observed in many diseases, particularly in cancer and in neurodevelopmental disorders, suggesting the importance of precise and carefully regulated miRNA dosage in the cells [4,5]. 

Maturation of most miRNAs occurs via the canonical pathway: transcription of an endogenous locus results in the primary-miRNA (pri-miRNA) containing one or more stem-loop structures that undergo two consecutive RNA cleavage reactions. The first occurs in the nucleus by the Microprocessor complex, a protein heterotrimer containing an RNase III enzyme, Drosha, and two molecules of its regulatory partner protein, the DiGeorge Syndrome Critical Region 8 (DGCR8) [6]. The so formed precursor-miRNAs (pre-miRNAs) are transferred to the cytoplasm by the Exportin-5 system, where another RNase III enzyme Dicer cleaves the apical loop, forming a double-stranded short RNA with two-nucleotide overhangs at their 3′ end. This intermediate then undergoes subsequent maturation steps when it is associated with an Ago protein and its auxiliary factors: one strand (the “passenger” strand) of the duplex is eliminated, and the final functional RISC is formed [7]. Certain miRNAs mature via alternative pathways during which one of the above cleavage steps (or both) is substituted by other unrelated processing mechanisms, such as the mRNA splicing for the Microprocessor-independent mirtrons, or other RNase activities for the tRNA-derived miRNAs [8]. 

Functionally linked miRNAs are often clustered in the human genome [9,10], and in the primate lineage, such clusters play essential roles in stem-cell regulation and in placenta formation [11,12]. The distinct regulation of these clusters goes beyond common transcriptional regulation, and details of the exact mechanisms often remain controversial [13]. The chromosome 19 miRNA cluster (C19MC) is an especially long (46 miRNA genes) cluster predominantly expressed in human embryonic stem cells (hESCs) and in the reproductive system. In hESCs, the position-dependent processing of this cluster is regulated by the Microprocessor recruitment, explaining why the miRNA expression levels in the Drosha knockdown of those cells gradually decrease toward the 3′ end of the cluster [14]. To further explore the molecular mechanisms behind this phenomenon, the recruitment and role of DGCR8 or additional Microprocessor-associated factors also need to be studied.

There are different approaches to uncover the regulation driven by miRNAs; knocking out DGCR8, Drosha, or Dicer in ES cells provides useful information about the biological processes affected by the general loss of miRNAs. For instance, global loss of DGCR8 and canonical miRNAs causes embryonic lethality in mice at d 6.5, and the balance between self-renewal and differentiation of DGCR8 knockout mESCs is severely disturbed [15,16]. In contrast, heterozygous deletion of DGCR8 in mES cells shows a cellular phenotype comparable with the wild type: the miRNA and DGCR8 mRNA levels are not significantly changed due to a feedback control by the Microprocessor complex [17]. Mice heterozygous for DGCR8 show reduced expression of DGCR8 and a subset of miRNAs in the prefrontal cortex. These effects are not observed in neonatal mice but emerge from postnatal days 25–30 [18]. DGCR8 deficiency also alters miRNA biogenesis in adult mouse hippocampus: results of the array-based miRNA analysis show that while most of the examined miRNAs are downregulated, some are increased, suggesting more complicated alterations in miRNA biogenesis than expected. In the subgranular zone of these monoallelic knockout mice, cell proliferation is reduced, which may affect adult neurogenesis in the hippocampal dentate gyrus [19]. In the forebrain, pyramidal neurons of DGCR8 +/− mice show decreased branch complexity and decreased total dendrite length. These neurons have altered electrical properties such as imbalance of spontaneous synaptic transmission and short-term plasticity [18,20]. In a human cellular model system of DiGeorge syndrome, cortical neurons derived from DGCR8 +/− hiPSCs show defected neuronal activity and calcium handling, and the monoallelic mutant cells have altered resting membrane potential and abnormal inactivation of voltage-gated calcium channels [21]. Here, we presented an in vitro model for investigating the heterozygous deletion of DGCR8 in hESCs. We studied the aspects of cell renewal capacity, as well as the differentiation pattern of these mutant stem cells, and addressed the questions of whether any DGCR8-related functional deficiency could be detected, among others, on the position-dependent regulation pattern of the C19MC.

## 2. Materials and Methods

### 2.1. Cell Culture and Differentiation

The HuES9 embryonic stem-cell line was kindly provided by Douglas Melton (HHMI). This work was performed according to ethical approvals (HuES9 NIH approval NIHhESC-09-0022 and Health Care Research Council, Human Reproduction Committee in Hungary (Egészségügyi Tudományos Tanács, Humán Reprodukciós Bizottság, ETT HRB) approval number 6681/2012-EHR. HuES9). The parental HuES9 and DGCR8-deficient HVRDe009-A-1 cells were maintained on hESC qualified Matrigel (Corning, New York, NY, USA, #354277) coated plates in mTeSR1 (STEMCELL Technologies, Vancouver, BC, Canada # 85850) with or without 0.8 µM puromycin (ThermoFisher Scientific, Waltham, MA, USA #A1113803). Media were changed every day. Cells were passaged with StemPro Accutase (ThermoFisher Scientific, Waltham, MA, USA #A1110501) at a 1:10 ratio every 3–4 days and plated onto fresh Matrigel-coated plates in mTeSR1-Y (mTeSR1 supplemented with 10 µM Y27632-2HCl (Selleckchem, Planegg, Germany, #S1049) for 24 h to improve cell survival. The genetic identity and normal karyotype of the cultured cells were confirmed by STR (short tandem repeat) analysis and G-Banding performed by UD-GenoMed Medical Genomic Technologies Ltd. For spontaneous differentiation, the hES cells were dissociated with ReLeSR (STEMCELL Technologies, #05872) and plated onto MEF (Merck, Darmstadt, Germany, #PMEF-CFL) covered tissue culture plates. Cells grown on MEFs were maintained in hESC culture media (KO-DMEM (ThermoFisher Scientific, #10829018) supplemented with 15% knockout serum replacement (ThermoFisher Scientific, Waltham, MA, USA, #10828010), 1 mM L-glutamine (ThermoFisher Scientific, Waltham, MA, USA, #25030024), 0.1 mM β-mercaptoethanol (ThermoFisher Scientific, Waltham, MA, USA, #21985023), 1% nonessential amino acids (ThermoFisher Scientific, Waltham, MA, USA, #11140050), and 4 ng/mL human basic fibroblast growth factor (ThermoFisher Scientific, Waltham, MA, USA, #13256-029)). hESC colonies were then dissociated with Collagenase (ThermoFisher Scientific, Waltham, MA, USA, #17018029) and cultured in suspension on low attachment plates in an EB (Embryoid body) medium (KO-DMEM supplemented with 20% FBS, 1 mM L-GLU, 1% non-essential amino acids, and 0.1 mM ß-mercaptoethanol) for 6 days. Next, EBs were transferred onto 0.1% gelatin coated 24-well tissue culture plates or confocal chamber slides (ThermoFisher Scientific, Waltham, MA, USA, #177402) and allowed to differentiate for another 6 days in DMEM (ThermoFisher Scientific, Waltham, MA, USA, #41965062) supplemented with 10% FBS. During differentiation, the media were changed every other day. The differentiated offspring was characterized by immunocytochemical staining and RT-qPCR.

### 2.2. Single-Cell Cloning

HVRDe009-A-1 hESCs were cultured without puromycin to enrich those cells not expressing the green fluorescent protein (GFP). Next, cells were dissociated with StemPro Accutase, and single cells were plated onto Matrigel-coated 96-well plates with a BD FACSAria™ Cell Sorter based on the GFP expression in cloning media (mTeSR1-Y supplemented with 1/3 MEF-conditioned hESC media). SSCs were expanded in cloning media for 10–20 days then replated onto 24-well Matrigel-coated plates in mTeSR1-Y. When reaching 80% density, SSCs were plated onto 6-well plates. GFP expression was validated by FACS measurements. 

### 2.3. Trichostatin A Treatment

Cells were plated onto 6-well plates and cultured without the addition of puromycin. On day 3, cells were treated with mTeSR1 supplemented with 30 nM or 60 nM Trichostatin A, respectively. Next day, cells were detached, and GFP expression was measured in the treated and untreated control cells with flow cytometry. In parallel, the expression of the endogenous cancer testis gene GAGE was measured via RT-qPCR to prove effective TSA-induced demethylation.

### 2.4. Flow Cytometry Measurements

SSEA-4 Flow Cytometry was performed as previously described [22]. Briefly, single-cell suspensions were prepared using Accutase. Subsequently, cells were labeled in PBS supplemented with 0.5% BSA (bovine serum albumin, Sigma, St. Louis, MO, USA, #A9418-50G) with anti-human SSEA-4 PE-conjugated antibody (1:100, R&D Systems, Minneapolis, MN, USA, #RD-FAB1435A-100) on 37 °C for 30 min. Propidium iodide (ThermoFisher Scientific, Waltham, MA, USA, #P1304MP) staining was employed for gating out the positively labeled dead cells. Control measurements with isotype-matched control were included (1:100, R&D Systems, Minneapolis, MN, USA, #IC007A).

### 2.5. Immunocytochemistry

For immunostaining, cells were seeded onto eight-well Nunc Lab-Tek II Chambered Coverglass (ThermoFisher Scientific, Waltham, MA, USA, #155411), fixed, and permeabilized with 4% PFA (paraformaldehyde) in DPBS (Dulbecco’s modified PBS) for 15 min on RT (room temperature). Next, cells were blocked for 60 min at RT in a blocking solution (DPBS supplemented with 2 mg/mL BSA, 0.1% Triton-X 100, and 1% fish gelatin with or without 5% goat serum). Then, the samples were incubated for 60 min at RT in the blocking solution supplemented with the following primary antibodies: OCT3-4 (1:50, Santa Cruz, CA, USA, #SC-5279) and NANOG (1:100, R&D Systems, Minneapolis, MN, USA, #AF1997) as pluripotency markers; AFP (1:500, Sigma, St. Louis, MO, USA#A8452), SMA (1:500, Abcam, Cambridge, UK, #ab7817), and ß-III-tubulin (1:2000, R&D Systems, Minneapolis, MN, USA, #RD-MAB1195) as markers specific for the three lineages. After washing with DPBS, the cells were incubated for 60 min at RT with Alexa Fluor 647-conjugated goat anti-mouse and Alexa Fluor 594-conjugated donkey anti-goat IgG antibodies (1:250, ThermoFisher Scientific, Waltham, MA, USA, #A11029, #A11012). DAPI (ThermoFisher Scientific, Waltham, MA, USA, # D1306) was used for nuclear staining. The samples were then examined by a Zeiss LSM 710 confocal laser scanning microscope.

### 2.6. RNA Isolation and Gene Expression Studies

Total RNA from the hES and differentiated cells was isolated using a TRIzol reagent (ThermoFisher Scientific, Waltham, MA, USA, #15596018). RNA integrity was analyzed by agarose gel electrophoresis; total RNA concentrations and sample purity were measured by a Nanodrop spectrophotometer (ThermoFisher Scientific, Waltham, MA, USA). 

For pri-miRNA analysis, 1 μg total RNA was reverse-transcribed by random oligomers using a High-Capacity cDNA Reverse Transcription Kit (ThermoFisher Scientific, Waltham, MA, USA). cDNA samples were diluted 1:10 before subsequent amplifications. In the case of C19MC pri-miRNA, RT-PCR was carried out by using a SYBR Green PCR Master Mix with custom-made PCR primers (Appendix A).

For gene expression assays, cDNA samples were synthesized from 400 ng of total RNA using the Promega Reverse Transcription system according to the manufacturer’s instructions. mRNA levels for DGCR8 (Hs00987085_m1), Drosha (Hs00203008_m1), NANOG (Hs02387400_g1), AFP (Hs00173490_m1), TBXT (Hs00610080_m1), and PAX6 (Hs00240871_m1) were determined using TaqMan^®^ gene expression assays (ThermoFisher Scientific, Waltham, MA, USA, #4331182).

Real-time PCR measurements were run and analyzed on the StepOne™ Real-Time PCR System (Applied Biosystems) according to the manufacturer’s instructions. Quantitative gene expression data were normalized to endogenous control mRNAs: for Taq-Man^®^ analyses RPLP0 (Hs9999902_m1) or PolR2A (Hs00172187_m1); for SYBR Green assay details, see Appendix A.

For mature miRNA quantification, the expression analysis was performed using the miRCURY LNA™ Universal RT miRNA PCR Assay (Qiagen, Venlo, Netherlands) according to the manufacturer’s instructions. Briefly, RNA samples (5 ng/µL) were reverse-transcribed, and the UniSp6 RNA spike-in template was added to each reaction for controlling the quality of cDNA synthesis. cDNA samples were diluted 1:80 before subsequent amplifications. RT-PCR was performed by using a miRCURY SYBR^®^ Green master mix (Qiagen Venlo, Netherlands), and real-time PCRs were run on a StepOnePlus™ platform (ThermoFisher Scientific, Waltham, MA, USA) according to the manufacturer’s protocol. In these cases, the hsa-miR-103a internal control miRNA was used for normalization during the relative quantifications by the ΔΔCt method.

### 2.7. Protein Analysis by Western blotting

Samples were briefly sonicated and subsequently ran on 8% acrylamide gels followed by electroblotting onto PVDF membranes. Membranes were then blocked with blocking solution (5% Milk/TBS-Tween) and incubated with a monoclonal antibody specific to human DGCR8 (1:1000, Abcam, Cambridge, UK, #ab191875) ON (overnight) at 4 °C. Next, anti-rabbit IgG (1:5000, ThermoFisher Scientific, Waltham, MA, USA, #G-21234) was used as a secondary antibody. Pierce ECL Western blotting substrate ((ThermoFisher Scientific, Waltham, MA, USA, # 32106) was used for signal detection; membranes were exposed to Agfa films. Anti-β actin antibody (1:10,000, Abcam, Cambridge, UK, #ab20272) was used to normalize the DGCR8 expression. Expression levels were determined by densitometry of the scanned images using ImageJ and corrected with background and normalized to β-actin and parental HuES9 levels. Briefly, a region of interest (ROI) for a given protein was chosen to be the smallest rectangle shape that can enclose the largest band of that protein and used in all lanes of a blot. The background for the normalization is measured by the same ROI at close proximity to the target band; aspecific bands (if any) were always avoided.

## 3. Results

### 3.1. Establishment of Heterozygous DGCR8 Mutant Clones

The heterozygous DGCR8 mutant HVRDe009-A-1 hESC line was established and characterized as previously described [23]. These mutant cells contain a donor DNA sequence in the third exon of DGCR8, which consists of two CAG-driven selection markers, a puromycin resistance gene, and a GFP. The insertion of the donor plasmid results in monoallelic DGCR8 expression in the mutant cells. The HVRDe009-A-1 hESCs show consistent GFP expression when cultured in a puromycin-containing medium; however, during puromycin deprivation, gradual loss of GFP expression was observed (Appendix A). Transgene silencing and mosaicism are known to occur in transfected cells, especially when selection is heavily dependent on antibiotic resistance [24,25]. Treatments with Trichostatin A (TSA), an organic compound interfering with the removal of acetyl groups from histones [26], resulted in elevated GFP expression only in the still GFP-positive cells, which suggests that gene silencing by deacetylation was not responsible for GFP loss (Appendix A). When sorting out GFP-positive and -negative populations, and propagated them without puromycin, GFP-negative cells showed a decreased GFP copy number based on real-time quantitative PCR measurements (Appendix A). These results show that the loss of transgene expression is likely a result of genetic rearrangements and the loss of the transgene copy from the DGCR8 locus. 

To investigate the exact genetic background in the GFP-positive and -negative cells, we accomplished single-cell cloning of the HVRDe008-A-1 cells based on the GFP expression. We established 12 GFP-positive (Appendix A) and 13 GFP-negative single-cell clones. The clones were propagated without puromycin, and the GFP expression in the clones was continuously monitored by high content screening and FACS measurements. By day 30, the GFP expression in 7 out of 12 GFP-positive single-cell clones decreased below 25% (Figure 1a), and by day 60, it further decreased to 0% in two clones (Appendix A); on the other hand, GFP-negative clones maintained to keep the negative phenotype, regardless of the remaining parts of the GFP expression cassette (GFP copy number was simultaneously monitored by quantitative real-time PCR measurements; see Figure 1b). 

### 3.2. Genetic Characterization of Selected Single-Cell Clones

Next, we have selected four single-cell clones for further characterization (Appendix A), namely HVRDe009-A-1-A11, HVRDe009-A-1-B3, HVRDe009-A-1-C4, and HVRDe009-A-1E9. All the investigated clones were negative for GFP expression (Figure 1a). Moreover, the single-cell clones lost their puromycin resistance and died after 48 h when treated with puromycin. 

The results prompted us to analyze the transgene sequence in these clones by diagnostic PCRs. Amplifying different segments of the transgene cassette, we found that significant portions of the GFP expression unit, as well as the puromycin resistance gene, were lost in these cells (Appendix A). However, certain regions of the transgene still remained integrated, indicating that at least one DGCR8 allele is disrupted in all of these descendant cells.

To confirm if one intact DGCR8 allele is still present, we sequenced the targeted genomic site in the clones, and Sanger sequencing data provided evidence for the monoallelic disruption of the DGCR8 gene (Appendix A). 

### 3.3. DGCR8 +/− hESCs Maintain Pluripotency and Trilineage Differentiation Capacity

The single-cell clones maintained stem-cell-like morphology and normal karyotype after single-cell cloning when cultured in feeder-free conditions (Appendix A). To evaluate the effect of the heterozygous DGCR8 mutation on the pluripotency, first we measured the SSEA-4 expression of the single-cell clones, and every clone showed over 90% positivity for this commonly used embryonic stem-cell marker (Figure 2a). Moreover, immunostaining of OCT4 and NANOG pluripotency markers showed homogenous expression in the hESC colonies (Figure 2b). 

To assess the differentiation capacity of these mutant hESCs, we performed an in vitro embryoid body (EB) differentiation assay. Immunostaining and real-time quantitative PCR measurements confirmed the continuous decline of the expression of pluripotency marker NANOG and increase in markers specific for the three germ layers (ectoderm: TUJ1/ β-III-Tub, PAX6; mesoderm: TBXT, SMA; endoderm: AFP) (Figure 3a,b). These results indicate that monoallelic deletion of DGCR8 does not cause impairments in the three-lineage differentiation capacity of the hESCs. 

### 3.4. Expression of Microprocessor Complex Components DGCR8 and Drosha

When determining the DGCR8 mRNA levels by real-time quantitative PCR, the clones showed slightly fluctuating but not considerably different levels when compared with their HuES9 stem-cell ancestor (homozygous to the wild-type DGCR8 allele). The Drosha mRNA levels also showed some variability among the clones when individually compared with their parental HuES9 cell line (Figure 4a and Appendix A). On the other hand, DGCR8 protein levels were decreased by at least 40–50% based on Western blot detections in each clone (Figure 4b,c and Appendix A). In contrast, Drosha protein levels considerably varied, showing rather clone-specific expression profiles (Figure 4b,c and Appendix A). 

### 3.5. Pri-miRNA Processing Efficiency of C19MC in DGCR8 Mutant Cells

The considerable decrease in DGCR8 protein levels prompted us to test whether the activity or any functions related to the Microprocessor complex are disturbed in the examined cell clones. In a recent study, we showed that the depletion of Drosha in hESCs caused a gradual decrease in pri-miRNA processing along an extended miRNA cluster, C19MC [14]. We also proposed that depleting DGCR8, the other component of the Microprocessor complex, may result in a similar phenotype; therefore, we tested the processing activity in three selected regions of C19MC in the four DGCR8 heterozygous clones. We detected clone-specific responses: a modest gradual decrease toward the 3′ end of the cluster was revealed in clones E9 and B3, whereas no significant decrease was measured in the other two clones, A11 and C4 (Figure 5). The results did not show a clear correlation with the measured DGCR8 expression levels but rather indicated a potential disturbance of the Microprocessor function among cells where the DGCR8 protein level is reduced due to a heterozygous mutation. 

## 4. Discussion

DGCR8 is a central component of the Microprocessor complex, and together with its endonuclease partner Drosha, they play an essential role in the initiating step of canonical miRNA biogenesis [6,27,28,29,30]. Disturbance in the level of DGCR8, therefore, significantly impairs miRNA maturation, negatively influencing cell proliferation, differentiation, and apoptosis [31,32], and thereby disturbing important developmental processes, including, among others, cardiovascular and brain development [33,34]. DGCR8 knockout or depletion strategies lead to identifying the functions of key miRNAs [35,36,37] but also revealed miRNA-unrelated, “noncanonical” functions of this protein, such as controlling the stability of small nucleolar RNAs [38,39] and its role in heterochromatin stabilization [40] or in DNA repair processes [41]. Considering its versatile cellular functions, it does not seem unexpected that the complete loss of DGCR8 is lethal in mice, and such knockout embryonic stem cells have serious differentiation problems [15]. On the other hand, mouse monoallelic DGCR8 mutant PSCs were reported to have no obvious phenotypes or changes in their differentiation capacity, and this was thought to be due to the homeostatic mechanisms affecting the DGCR8 mRNA levels [17,42,43]. However, subsequent studies using these mouse models carrying monoallelic mutations revealed pronounced downstream effects in neurological physiology, manifested in reduced cell proliferation and aberrant neuron morphology in several brain areas [18,19,20], or in cardiac malfunctioning, resulting in heart failure [44]. Moreover, conditional mouse knockout models in early immune cells resulted in defects in natural killer cell activation and survival, which was connected to the loss of certain miRNA population [35]. In a human cellular model system of DiGeorge syndrome, cortical neurons derived from DGCR8 +/− hiPSCs showed defective neuronal activity and calcium handling, which were in line with the results of the mouse models [21]. However, DGCR8 heterozygous deficiency is still poorly characterized in human ESCs and in differentiated tissue types, mostly due to the lack of suitable knockout or mutant model cell lines.

To investigate the function of the DGCR8 in various human cell types, we previously established a monoallelic mutant hESC-line using the CRISPR/Cas9 system [23]. This is a suitable model to study the molecular and cellular defects not only in stem cells but also in their differentiated isogenic offspring cell types. However, when generating several single-cell clones for further comparative studies, and omitting the antibiotic selection to avoid interference with differentiation, we detected a gradual loss of the inserted transgenes, the GFP marker and the puromycin resistance gene. After the genetic analyses of the clones, we concluded that it is not due to any form of epigenetic silencing but rather to a recombination event initiated by the two CAG promoter sequences present in the transgenic cassette, leading to the disruption of the transgene structure and to the loss of expression (Appendix A). Such transgenic rearrangements and somatic drifts are known problems reported earlier when establishing stable genetic models in several systems [45,46,47,48]; however, as we successfully applied similar expression cassettes with two transcription units in earlier studies, even in stem cells [49,50,51], we did not expect this form of transgene inactivation. Nevertheless, the modified DGCR8 allele still remained mutant in all clones even after the transgene rearrangement, so these cells could be used as DGCR8 monoallelic mutants for further studies.

The effect of DGCR8 haploinsufficiency on cellular phenotypes is controversial: such cells are viable, but since the impairment of this regulatory protein significantly varies for different miRNAs [52], the effect on the functional disturbance is also cell-type-specific. In our investigations at the human stem-cell level, the DGCR8 monoallelic mutation did not disturb either the pluripotency status or the tri-lineage differentiation capacity, the results of which were in line with the mouse models [17,18]. In addition, measuring the mRNA expression levels did not reveal dramatic differences between the normal and heterozygous mutant cell lines, which may have been expected based on the subtle autoregulatory mechanism described earlier for this transcript [17,42,43]. In contrast, studies at the protein level revealed significant decrease in DGCR8 expression as compared with the normal hESCs. This result prompted us to investigate genetic loci where such an imbalance in Microprocessor components is known to disturb molecular functions. One candidate was the C19MC locus, where the local depletion of the Microprocessor complex results in a gradual positional decline of pri-miRNA processing in a long miRNA cluster. This positional effect was more prominent when miRNA positions were compared in a pairwise manner between hESCs and placenta cells, between cell types where this imprinted miRNA cluster is predominantly expressed [14]. When we investigated the single-cell clones carrying the DGCR8 heterozygous mutation, the positional effect was detected in only half of the examined clones, indicating that the significantly lower protein level can result in a partial functional disturbance of the Microprocessor complex in hESCs. However, it is currently unknown why this heterozygous mutation did not show a 100% penetrance in our experiments. One explanation could lie in the naïve versus primed stage of the used human embryonic stem cell line: in a recent study, it was revealed that the C19MC is relatively highly expressed in naïve hESCs, whereas its transcriptional activity is declined in primed stem cells [53]. In our culturing conditions, the HuES9 ESC line rather contains primed cell populations, and the lower expression of the miRNA cluster in those cell types might hinder the detection of the position effect in pri-miRNA processing. In addition, C19MC is also expressed at a much higher level in the trophoblast lineage, a differentiation route that is more effectively initiated from the naïve state of hESCs. It is also conceivable that the effect of the DGCR8 monoallelic mutation may be more pronounced in those cell types. However, further experiments, including RNA sequencing, are needed to uncover the DGCR8 mRNA profile and the potential presence of nontranslatable splicing variants in our established clones, to unequivocally connect all the observed functional changes to the DGCR8 protein itself.

In conclusion, in this study, we could show that the progenies of the hESC line carrying a heterozygous mutation in the DGRC8 gene are good molecular and cellular models to examine the functions of this RNA-binding regulatory protein. Moreover, being the good basis to generate isogenic differentiation lineages, they provide an excellent in vitro platform to study the phenotype of DGCR8 deficiency in several human somatic cell types. Since one consequence of lower DGCR8 expression is the suppressed maturation of canonical miRNAs, the use of our special hESC line can also enhance the studies of mirtrons, a special class of Microprocessor-independent but splicing-dependent miRNAs [54,55,56,57,58,59]. Taken together, this DGCR8 +/− cell line can well-contribute to the study of the dominant human RNA interference pathway, as well as to the better understanding of the “noncanonical” functions of the DGCR8 protein. 

## Figures and Tables

**Figure 1 genes-13-01925-f001:**
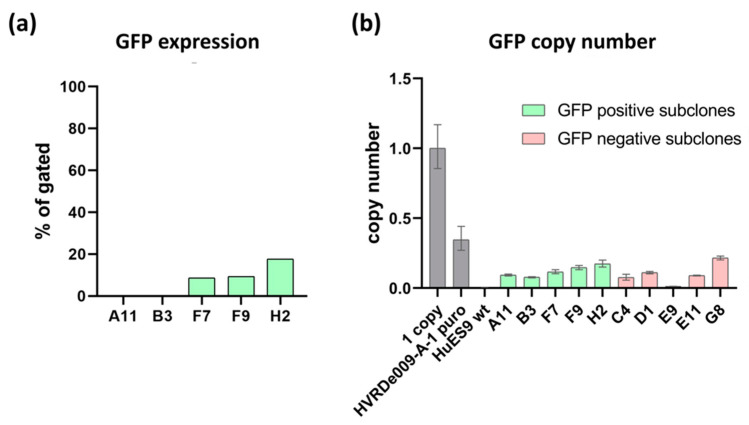
GFP expression in the single-cell clones of HVRDe009-A-1. (**a**) GFP FACS measurements of GFP-positive single-cell-derived clones (passage 10 after single-cell cloning); (**b**) GFP copy number measurements on the GFP-positive and -negative single-cell-derived clones and the HRDVe009-A-1 cells cultured with puromycin. Relative quantitation of copy number values was calculated using RPPH1 as a reference target and 1 copy control gDNA reference sample.

**Figure 2 genes-13-01925-f002:**
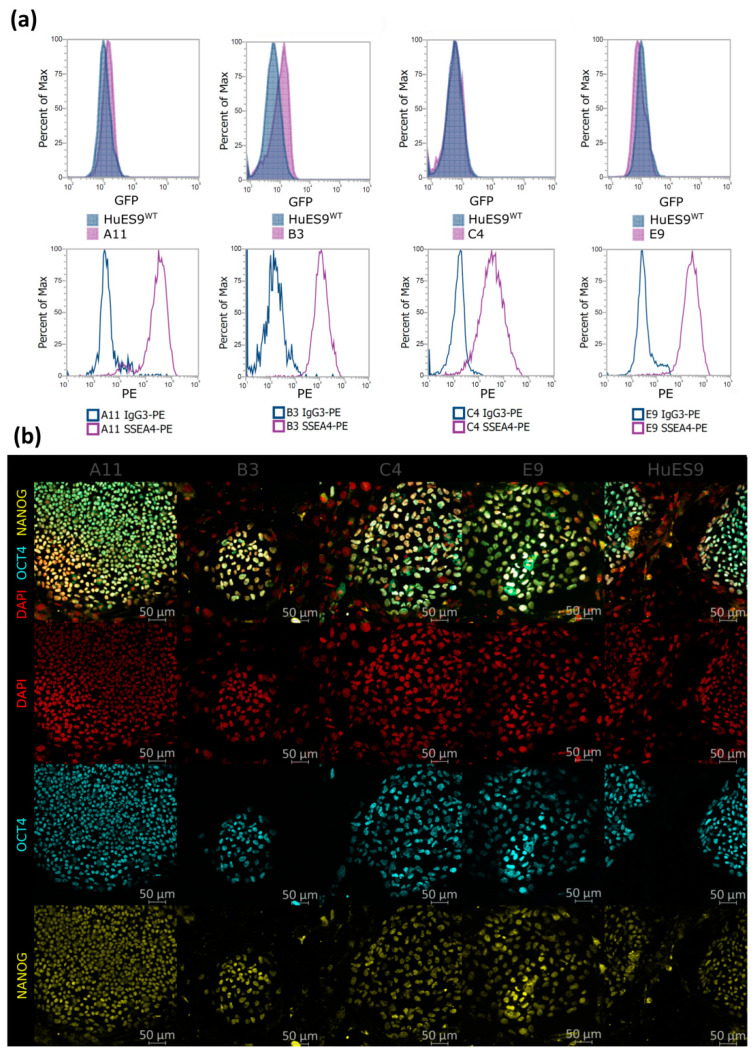
Pluripotency of the monoallelic mutant single-cell clones. (**a**) GFP and SSEA-4 FACS measurements of undifferentiated hESCs. (**b**) Immunostaining of the pluripotency markers OCT4 and NANOG on undifferentiated hESCs.

**Figure 3 genes-13-01925-f003:**
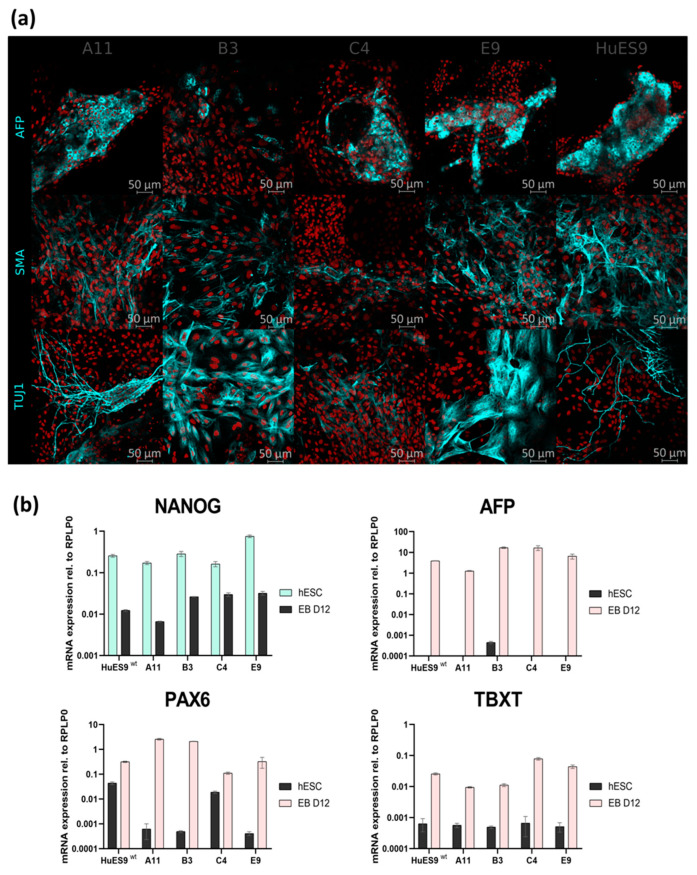
Differentiation capacity of monoallelic mutant single-cell clones. (**a**) Immunostaining of markers specific for the three germ layers (AFP, SMA, TUJ1) on the differentiated offspring (embryoid body day 12). (**b**) mRNA expression levels of pluripotency and lineage-specific markers on undifferentiated hESCs and differentiated offspring (embryoid body day 12).

**Figure 4 genes-13-01925-f004:**
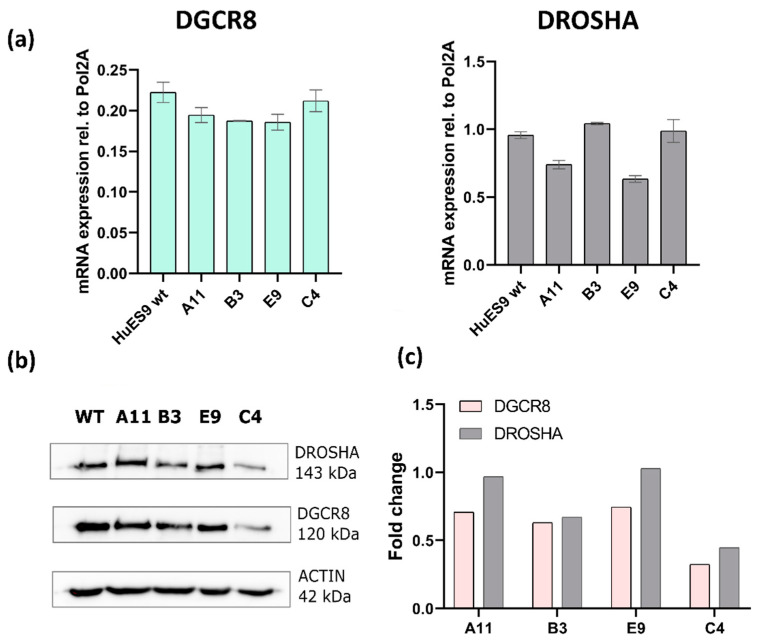
Expression of the Microprocessor components DGCR8 and Drosha in the monoallelic mutant single-cell clones. (**a**) Expression levels of DGCR8 and Drosha mRNAs relative to POL2A endogenous control in a representative experiment. Data are presented as mean ± SD of 3 technical parallels. (**b**) Representative Western blots for DGCR8, Drosha, and β-actin. (**c**) DGCR8 and Drosha protein levels of the representative Western blots normalized to β-actin levels, and to the relative expressions in the parental HuES9 (WT) cell line.

**Figure 5 genes-13-01925-f005:**
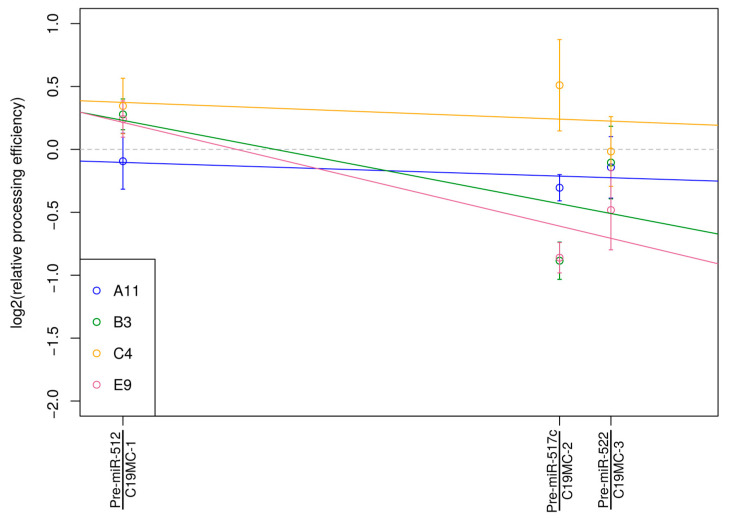
Measuring position-dependent pri-mRNA processing along the C19MC in DGCR8 mutant clones. At 3 selected positions, the ratio of total versus unprocessed pri-miRNAs was determined using distinct primer pairs by real-time PCRs. In this representative experiment, colored circles show mean values of measurement points in different clones; ± SE values of 3 technical parallels are also shown. Colored lines indicate the tendency of decrease in processing efficiency (if any) for a given clone.

## Data Availability

The data presented in this study are available on request from the corresponding authors.

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
