# Peer review of "Partial Disturbance of Microprocessor Function in Human Stem Cells Carrying a Heterozygous Mutation in the DGCR8 Gene"

_genes, 2022, doi:10.3390/genes13111925_

Round 1
Reviewer 1 Report
DiGeorge Syndrome Critical Region 8 (DGCR8) protein is involved in maturation of microRNAs as a part of Microprocessor complex. The authors established a human embryonic stem cell (hESC) line carrying a monoallelic DGCR8 mutation by using the CRISPR-Cas9 method. The results did not show a clear correlation with the measured DGCR8 expression levels but indicated a potential disturbance of Microprocessor functions among cells with lower level of DGCR8 protein. The work is supported by many main and supplementary data and limitations were also discussed.
Please, correct carefully the text according to small editorial problems, e.g.
1. line 51 - unnecessary hyphen in "Di-George Syndrome"
2. line 77 - unnecessary hyphen in "be-tween"
3. line 302 - unnecessary hyphen in "re-al-time"
4. line 120 - "Scientific, #10829018) supplemented with XXX)." XXX should be replaced with the proper information.
Author Response
>1. line 51 - unnecessary hyphen in "Di-George Syndrome"
>2. line 77 - unnecessary hyphen in "be-tween"
>3. line 302 - unnecessary hyphen in "re-al-time"
Thank you for pointing these out, the typos have been corrected.
>4. line 120 - "Scientific, #10829018) supplemented with XXX)."
XXX should be replaced with the proper information.
This was a clear mistake in the text, the ‘XXX’ has been replaced with the appropriate information.
We appreciate the positive assessment of our manuscript and we hope that the Reviewer finds the revised version acceptable for publication.

Reviewer 2 Report
Authors in their paper analyze heterozygous human embryonic stem cells with monoallelic Dgcr8 expression. Such cells could be used for analysis of Dgcr8 function(s) dependent and independent on miRNA processing.
Comments:
1) Characterization and quantification of Dgcr8 and Drosha protein levels (Fig. 4) is not convincing. I have concerns regarding the quantification procedure: how many experiments were used for Fig. 4b calculations? Are results from a typical WB experiment shown as no error bars are presented? Personally, I do not see strong difference in expression on WB presented in Fig. 4c – also the bands are of irregular shape which should be quite difficult to use for quantification… As this result is quite important for conclusions of the paper, I would really suggest to focus on results of these experiments.
Also the interpretation (lines 300-307) is questionable: claims “...no significant changes in DROSHA…” and “...DGCR8 mRNA levels showed more variability among the clones…” are supported by the presented data (Fig. 4a). Is it swapped (as Drosha seems to be more variable for me)? Claims “…DGCR8 levels were 305 decreased by at least 40-50%…“ and “…Drosha protein 306 levels varied considerably…” are also not supported by Fig. 4b (seems to me to behave very similarly).
What is the difference between Fig. 4a and Fig. S7? The results seem to be quite different…
Overall, I would suggest to thoroughly revise this part and rephrase the corresponding text.
2) Functional analysis in Fig. 5 is not much convincing in the presented form. No error bars are presented – how many experiments/repetitions were done?
3) In my opinion, there is not much difference in Dgcr8 expression in clones analyzed. Authors sequenced disrupted Dgcr8 allele to confirm monoallelic expression. But, was similar analysis performed also on RNA level? Theoretically, it is possible that disrupted allele is still expressed with some sort of alternative splicing and produce alternative/shorter Dgcr8 isoforms – as it is not clear which part of protein is recognized by antibody used and some faint bands are present on WB gels (Fig. 4c)… Such potential isoforms could influence the overall functional outcome.
4) Did authors try some alternative methods to disrupt Dgcr8 allele and down-regulate its expression? Such as “simple” CRISPR-based deletion of a region of interest (e.g. whole gene including a promoter), transfection of siRNAs, expression of shRNAs (as these are usually differentially active and could down-regulate Dgcr8 expression to variable levels)?
5) Conclusions (lines 415-418) are rather exaggerated – more profound characterization (including also RNA-seq and similar methods) would be needed to fully characterize the clones and connect observed functional changes unequivocally to Dgcr8 protein itself. I would suggest to slightly rephrase this section.
Minor points/typos:
-line 120: information is missing (XXX)
-lines 232-233 seem to me redundant (similar to lines 221-222)
-line 243: claim “…while GFP negative clones maintained to keep the negative phenotype (Figure 1.a)” is not clear for me how is it illustrated in the figure 1a
-line 248: use “passage 10” instead of “P10“ to avoid misunderstanding with a name of the clone
-Figure S5: it is not clear to me which part of Dgcr8 allele was sequenced: I would welcome a scheme (similar to Fig. S4) indicating the localization of primer(s) used for sequencing
-check hyphens in lines 54, 77, 132, 302, 389
-line 63: delete a bracket
-line 226: … resulted in elevated …
-line 227: word “originally” is not clear for me – should be rather “still” or something of this meaning?
-correct format (bold font) in Fig. 2 legend to be similar to other figures
-line 298: a parenthesis is missing (after “12”)
-line 312: a parenthesis is missing (after “b”)
